# Assessment of antenatal care quality in Ethiopia: Facility-based study using service provision assessment data

Addisu Alehegn Alemu[1,2]*, Alec Welsh[3,4], Theodros Getachew[5,6], Marjan Khajehei[2,7,8,9]

**1** College of Health Sciences, Debre Markos University, Debre Markos, Ethiopia, **2** School of Women's and Children's Health, University of New South Wales, Sydney, Australia, **3** Discipline of Women's Health, School of Clinical Medicine, University of New South Wales, Sydney, Australia, **4** Department of Maternal-Fetal Medicine, Royal Hospital for Women, Sydney, Australia, **5** Health System Research Directorate, Ethiopian Public Health Institute, Addis Ababa, Ethiopia, **6** Department of Global Health and Population, Harvard T.H. Chan School of Public Health, Takemi Program in International Health, Boston, Massachusetts, United States of America, **7** Women's and Newborn Health, Westmead Hospital, Sydney, Australia, **8** The University of Sydney, Sydney, Australia, **9** Western Sydney University, Sydney, Australia

* addisua4@gmail.com

**Data Availability Statement:** The ESPA data relevant to this study are available from Demographic and Health Surveys (DHS) (https://www.dhsprogram.com/data/dataset/Ethiopia_

## Abstract

### Background

Antenatal care (ANC) coverage in low- and middle-income countries has increased in the past few decades. However, merely increasing care coverage may not enhance maternal and newborn health unless the recommended service components are also provided. Our aim was to assess the quality of ANC and its associated factors in Ethiopia.

### Methods

We used data from 2,042 pregnant women whose first ANC consultation was observed. Data were obtained from the 2021–2022 Ethiopian Service Provision Assessment survey conducted among a nationally representative sample of 1,158 healthcare facilities. Twenty-four components of ANC were assessed, and their values were summed to generate a total ANC quality score (range: 0–24). A higher score indicated a superior ANC quality. A multiple generalized Poisson regression model was fitted to identify factors influencing the quality of ANC. All statistical analyses were performed using STATA version 16.

### Results

The mean ANC quality score was 11 (standard deviation [SD]: 3.8). Blood pressure measurement was the most commonly performed ANC component, at 79.5%, and breast examination for cancer screening was the most neglected component of ANC, at 3.3%. ANC quality was higher in the Amhara region (incidence rate ratio [IRR]: 1.088; 95% confidence interval [CI]: 1.0–1.171) and Southern Nations, Nationalities, and Peoples' Region (IRR: 1.081; 95% CI: 1.002–1.166), and when the care was provided by a female healthcare worker (IRR: 1.054; 95% CI: 1.021–1.088). On the other hand, ANC quality decreased in rural healthcare facilities (IRR: 0.964; 95% CI: 0.932–0.998), clinics (IRR: 0.666; 95% CI:

SPA_2021.cfm?flag=0). Researchers must register in order to access data. The authors confirm that others would be able to access these data in the same manner as them and that the authors did not have any special access privileges that others would not have.

**Funding:** The author(s) received no specific funding for this work.

**Competing interests:** The authors have declared that no competing interests exist.

**Abbreviations:** ANC, Antenatal Care; CI, Confidence Interval; ESPA, Ethiopian Service Provision Assessment; GPR, Generalized Poisson Regression; IRR, Incidence Risk Ratio; MEASURE DHS, Monitoring and Evaluation to Assess and Use Results Demographic and Health Surveys; SNNP, South Nations, Nationalities, and Peoples regions; SE, Standard Error; WHO, World Health Organization.

0.581–0.764), and health posts (IRR: 0.817; 95% CI: 0.732–0.91). Similarly, ANC quality decreased when gestational age at the first antenatal visit increased (IRR: 0.994; 95% CI: 0.992–0.996) and when care was received from a non-nearby healthcare facility (IRR: 0.956; 95% CI: 0.923–0.990).

## Conclusion

Overall, the quality of ANC in Ethiopia is suboptimal. Encouraging women to initiate ANC early and utilize nearby facilities, assisting providers in delivering standardized services through preservice training, supervision, and continuous education, and ensuring the availability and proper use of necessary resources at all facilities are important to improve ANC quality.

## Introduction

Despite global efforts to reduce maternal and newborn deaths under the Sustainable Development Goals [1], studies conducted five years later show that progress has either stagnated or declined in nearly all United Nations regions [2]. In 2020, a total of 287,000 maternal deaths and 2.4 million neonatal deaths were reported globally [3,4]. Almost 70% of maternal deaths and 41% of neonatal deaths were reported in Sub-Saharan Africa [4,5]. The leading causes of maternal death were obstetric hemorrhage, hypertensive disorders in pregnancy, non-obstetric complications, and pregnancy-related infections [6,7]. The leading causes of neonatal mortality were identified as perinatal asphyxia, severe neonatal infections, and preterm birth [7]. Most maternal and neonatal deaths can be avoided with access to quality care throughout pregnancy and during and after childbirth [8].

Antenatal care (ANC) is medical care given to pregnant women to ensure the best health outcomes for both mothers and unborn babies [9]. ANC enables health promotion, disease prevention, and early diagnosis and treatment of pregnancy-associated complications, thereby helping to prevent maternal and neonatal deaths [9,10]. As the burden of both communicable and noncommunicable diseases increases globally, the importance of ANC is growing. It provides an opportunity to screen, diagnose, and manage pregnancy-associated hypertension, diabetes, and pre-eclampsia [11]. It also provides a valuable opportunity for effective screening and management of common infectious diseases including malaria [12], hepatitis [13], syphilis [14] and Human Immunodeficiency Virus/Acquired Immunodeficiency Syndrome (HIV/AIDS) [15]. It also provides women with guidance on safe childbirth, newborn care, breastfeeding, postnatal recovery, and family planning [16].

The 2016 World Health Organization (WHO) ANC guidelines recommend at least eight ANC visit episodes with qualified healthcare providers to ensure optimal ANC, with specific content and interventions outlined for each visit [9]. There are no universally accepted quality healthcare metrics [17–21], but monitoring the level of adherence to the recommended content and interventions during each ANC visit can provide an assessment of the overall quality of ANC [22]. The first visit episode, ideally within the first 12 weeks of gestation, covers a broad, detailed spectrum of services, including medical history collection, pregnancy confirmation, physical examination, general and targeted screenings, and the provision of key interventions. Subsequent contact episodes focus on re-evaluation of changes from the previous condition and detection of any new developments [9].

Ethiopia is one of the Sub-Saharan Africa countries with high maternal and neonatal death rates, with 412 maternal deaths per 100,000 live births and 29 neonatal deaths per 1,000 live

births [23]. While there has been significant improvement in the number of women attending at least one ANC visit in Ethiopia [24], issues persist with timely visits [25], meeting the recommended number of visits [24], and using all recommended service components [26]. The 'three delays' model provides a clear explanation of the main challenges women encounter when seeking maternal health services [27]. The model identifies the barriers to accessing maternal health services across three interrelated levels before maternal and/or neonatal death occurs. First, women may delay seeking ANC due to low social status, lack of awareness about complications, previous poor care experiences, traditional practices, normalization of maternal death, and financial dependency. Studies in Ethiopia show that the knowledge pregnant women have about pregnancy-associated danger signs is suboptimal [28,29]. This may influence their decisions regarding the use of maternal care, including ANC. Second, delay in reaching a health facility may result from factors such as distance, lack of infrastructure (such as roads or transportation), and challenging terrain. Third, delays in receiving adequate quality care may be due to a shortage of properly trained health staff and the unavailability of medical supplies and equipment.

Previous studies in Ethiopia have assessed ANC quality by evaluating the timing of service initiation, the number of ANC visits per pregnancy, the number of service components, and the level of client satisfaction toward the service [26,30]. However, they have failed to evaluate ANC quality by observing the service provided to pregnant women, which is considered the gold standard for assessing care quality [31]. Evaluating the quality of ANC at the healthcare facility level, based on data from direct clinical consultations, is an effective way to precisely identify the level of ANC quality [32,33]. After consideration of the gaps in the literature, we aimed to assess the quality of ANC and associated factors in Ethiopia by analyzing the 2021–2022 Ethiopian Service Provision Assessment (ESPA) survey data. The survey is the second in Ethiopia to assess the capacity and potential of healthcare facilities to deliver quality healthcare nationwide [34].

This study employed a conceptual framework adapted from the SPA quality of care framework, guided by the Demographic and Health Surveys Program. The framework aims to provide a comprehensive evaluation of quality care within a facility, focusing on two interrelated components of quality: structure and process. Structural quality refers to the availability of human and physical resources, while process quality pertains to how well clients receive care that adheres to established standards, including effective communication between providers and clients, as well as a focus on client-centered care [35]. As structural quality reflects health facilities' readiness to deliver quality care [36], this study focused on the process aspects of quality care.

## Methods

### Data source

We used the 2021–2022 ESPA survey data. The survey was conducted from 11 August 2021 to 4 February 2022. The Ethiopian Public Health Institute received funding from the United States Agency for International Development and the Government of Ethiopia, and conducted the survey in collaboration with the Ethiopian Ministry of Health [34].

### Settings

Ethiopia is a country in eastern Africa and constitutes the majority of the Horn of Africa's landmass. Ethiopia is geographically structured into 10 regional states: Afar, Amhara, Oromia, Somali, Benishangul-Gumuz, Gambela, Tigray, Southern Nations Nationalities and People's Region, Harari, and Sidama. There are two administrative cities (Addis Ababa and Dire Dawa)

[37]. A total of 409 hospitals, 3,713 health centres, 5,572 private clinics, and about 17,654 health posts serve the people of Ethiopia. According to the Ministry of Health's 2021–22 Annual Performance Report, there is an insufficient number of healthcare professionals across all types to meet the Sustainable Development Goals threshold. For example, there are 1.3 physicians (general practitioners, specialists, subspecialists, and dental surgeons) per 10,000 people, and 7.1 nurses per 10,000 people [34].

## Sampling

The 2021–2022 ESPA was conducted among multistage, stratified, randomly selected healthcare facilities in nine regions and the two administrative cities of the country, excluding the Tigray region. The first stratification involved health facilities by region and type. Following that, clinics in each region underwent additional stratification based on their classification (higher, medium, lower, and specialty clinics), with power allocation taken into account to ensure comparable survey precision across included regions and cities. All hospitals in the country were included due to their relatively small number and significance, while health centers were proportionally sampled across regions except for Dire Dawa and Harari, where all health centers were included due to their small number. Similarly, health posts were sampled with power allocation taken into account across the included regions and cities. Based on this, the final sample size of healthcare facilities included for the survey was 1,407. However, only 1,158 healthcare facilities were surveyed, excluding those in Tigray due to security issues and facilities that were closed or converted to COVID-19 care centers. Out of the total number of institutions surveyed, only 905 offered ANC, in which a total of 4,355 pregnant women were observed while receiving their ANC service and interviewed immediately after their consultations. Pregnant women at healthcare facilities for ANC were systematically sampled based on daily service demand, with a maximum of 15 selected per facility [34]. We analyzed data from 2,042 pregnant women who attended their first ANC visit in the healthcare facilities (Fig 1). This is due to the fact that the interventions and procedures recommended for pregnant women differed based on the number of ANC visits, with the first visit encompassing a wider and more comprehensive range of services compared to the following visits [9].

## Data collection procedure

The ESPA employed three fundamental questionnaires that have been adapted from the Monitoring and Evaluation to Assess and Use Results Demographic and Health Surveys (MEASURE DHS) project. These questionnaires include the facility inventory, healthcare provider, client exit questionnaire, and observation checklist. The tools were first prepared in English and then translated into the three main languages of the country: Amharic, Oromiffa, and Tigrigna. Data were collected electronically using the Census and Survey Processing System software. A pre-test was carried out before the data collection period, and all data collectors, supervisors, and regional coordinators participating in the fieldwork were trained [34]. In our study, the data from the pregnant women were linked with the data of their care providers and healthcare facilities to assess the influence of facility- and provider-related factors.

**Main outcome variable.**  The main outcome variable for this study was quality of ANC.

**Main outcome measure.**  Measuring the quality of ANC is a complex task as not all aspects of quality can be easily measured [38]. However, the number of comprehensive recommended components of ANC provided can be an indicator of the quality of the service [39]. Thus, ANC quality indices were constructed based on the components of the service outlined under the WHO Service Availability and Readiness Assessment manual [40], and the Ethiopian National Antenatal Care Guideline [41]. We therefore considered 24 ANC components

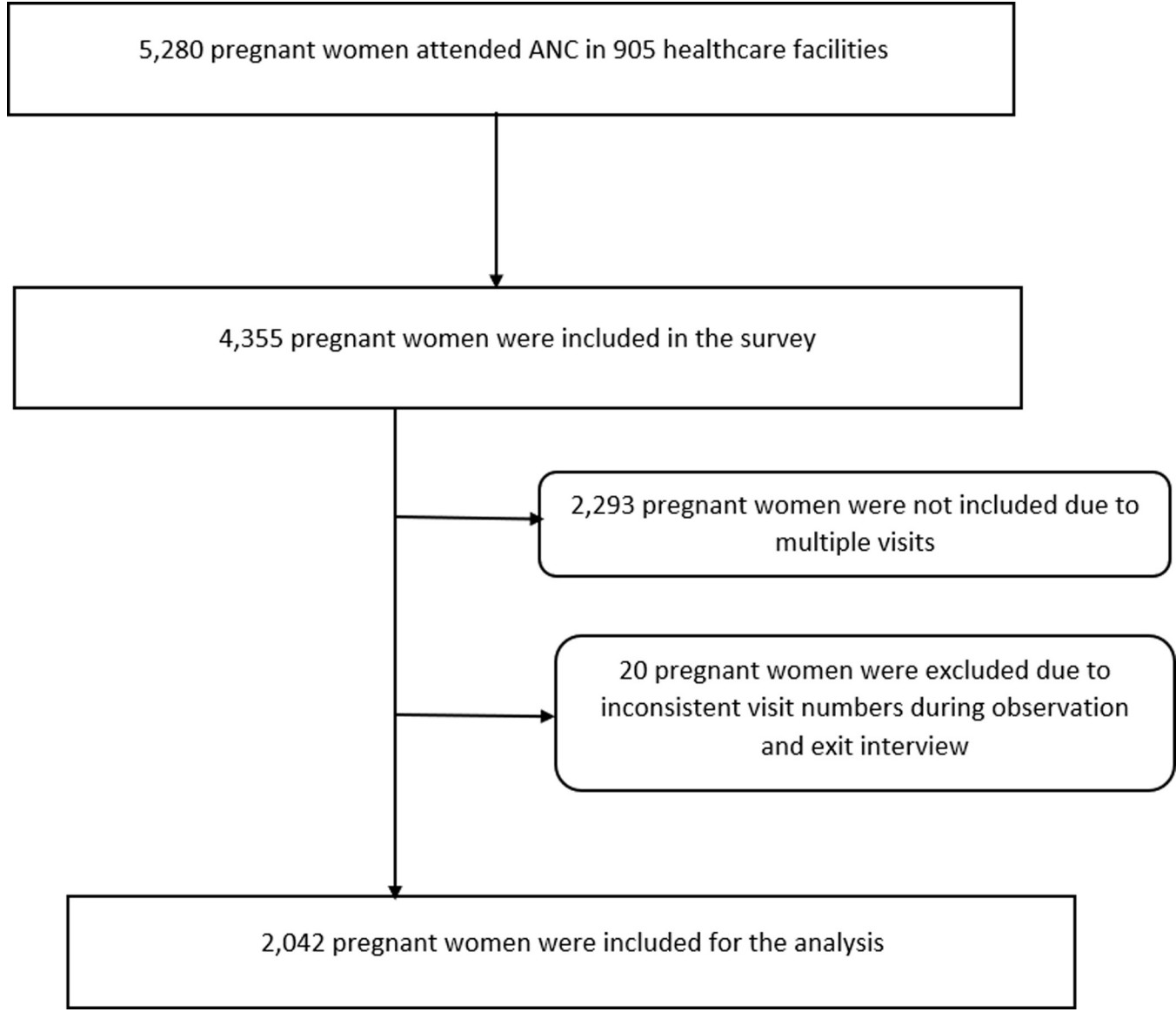

**Fig 1. Study flow chart extracted from the 2021–2022 ESPA dataset.**

(Table 2) that were found in the 2021–2022 ESPA dataset and were expected to be provided to every pregnant woman during her initial ANC visit. Each component was assigned a score of 1 when the care was provided and a score of 0 when the woman did not receive that specific care component. The total score ranged from 0 to 24, with a higher score indicating better quality of ANC [33,42]. The selection of the components aligned with related literature from other studies [43,44].

**Independent variables.** Through a comprehensive review of the literature, we identified factors that could determine the quality of ANC. These determinant factors were categorized into women-related factors (i.e., the woman's age, marital status, weeks of pregnancy at first antenatal visit, area of residence, educational status, gravidity, health insurance coverage, and her partner's involvement during ANC); healthcare provider-related factors (gender and profession); and facility-related factors (region, facility ownership, distance from home, facility

type and location) [36,45,46]. Detailed descriptions of the independent variables can be found in (S1 Table).

## Data management and analysis

The ESPA service components were scored as being either provided or not [34]. The data were collected, entered, edited, and cleaned. We analyzed only the available data rather than adopting an alternative technique for managing missing data [47]. We conducted data analysis using STATA (Version 16; StataCorp, College Station, TX). Since the outcome variable (ANC quality) was a non-negative integer, we employed the Poisson regression model. However, the standard Poisson regression assumes equal mean and variance, a condition known as equidispersion [48]. The mean and variance of the observations for our study, which were 10.99 and 3.84, respectively, indicate that the assumption of Poisson regression was violated due to underdispersion. Therefore, we fitted the most effective model, which was an extension of Poisson regression called generalized Poisson regression (GPR), to obtain precise results [48]. We analyzed the unweighted data because weighting data for Poisson regression would affect regression estimates [49]. We summarized the characteristics of the study participants and conducted both bivariable and multivariable GPR to assess the relationships between the ANC quality and the independent variables. All independent variables that exhibited a p-value below 0.20 in the binary GPR analysis were incorporated into the multivariable GPR model. However, the significance level in the multivariable GPR was determined at a p-value of less than 0.05. The results from both the binary and multivariable GPR models were presented using incidence rate ratios (IRRs) with 95% confidence intervals (CIs).

## Ethics approval and consent to participate

We received ethics clearance from both the Demographic and Health Survey Program in Ethiopia, and the Human Research Ethics Committee at the University of New South Wales in Sydney. All procedures adhered to the protocol outlined in the ESPA survey. As the data were anonymized, obtaining informed consent from the participants was not required.

## Results

### Characteristics of the study participants

In this study, records of 2,042 pregnant women who visited a healthcare facility for ANC were analyzed. The mean age of the participants was 25.3 years (SD: 4.9 years). Two-thirds of the participants (66.2%) lived in urban areas. Most of them (97.5%) were married. Nearly one-quarter of them had no formal education, and only 308 (14.8%) of the participants were educated at a college level or above. Only 23.5% of the participants had a prepayment plan or insurance that paid part of the cost of their healthcare services, and just over half of them (53.8%) were not accompanied by their partners during their ANC visits (Table 1).

### Quality of ANC

The mean score for ANC quality was 11.0 (SD: 3.8). The most commonly provided components of service were blood pressure measurement (79.5%), asking for the date of last menstruation (77.3%), and measuring weight (75.7%). Conversely, examining the breast for masses (3.3%), examination for swollen lymph nodes (5%), and examining extremities for edema (8.2%) were the most commonly neglected components of the service (Table 2).

**Table 1. Characteristics of the study participants (n = 2,042).**

| Variable | Category | Number (%) or mean ± SD |
|---|---|---|
| Maternal age (years) (n = 2,023) | Range 12–49 | 25.3 ± 4.9 |
| Gestational age at first antenatal visit (weeks) (n = 2,019) | Range 4–44 | 22.0 ± 8 |
| Gravidity (n = 2,042) | Primigravida | 677 (33.1%) |
| | Multigravida | 1,365 (66.9%) |
| Marital status (n = 2,042) | Unmarried | 51 (2.5%) |
| | Married | 1,991 (97.5%) |
| Maternal education status (n = 2,038) | No education | 472 (23.1%) |
| | Primary school | 689 (33.8%) |
| | Secondary school | 576 (28.3%) |
| | College and above | 301 (14.8%) |
| Area of residence (n = 2,042) | Urban | 1,352 (66.2%) |
| | Rural | 690 (33.8%) |
| Region (n = 2,042) | Addis Ababa | 126 (6.2%) |
| | Amhara | 375 (18.4%) |
| | Oromia | 744 (36.4%) |
| | Somali | 139 (6.8%) |
| | SNNP[a] | 337(16.5%) |
| | Sidama | 130 (6.4%) |
| | Other[b] | 191 (9.3%) |
| Qualification of the care provider (n = 2,042) | Medical doctor | 285 (13.9%) |
| | Health officer | 155 (7.6%) |
| | Nurse | 817 (40.0%) |
| | Midwife | 418 (20.5%) |
| | Laboratory professional | 208 (10.2%) |
| | Other[c] | 159 (7.8%) |
| Partner's involvement (n = 2,041) | No | 1,098 (53.8%) |
| | Yes | 943 (46.2%) |
| Healthcare facility nearest to home (n = 2,041) | Yes | 1,435 (70.3%) |
| | No | 606 (29.7%) |
| Have healthcare insurance (n = 2,041) | Yes | 480 (23.5%) |
| | No | 1,561 (76.5%) |
| Care provider's gender (n = 2,042) | Male | 847 (41.5%) |
| | Female | 1,195 (58.5%) |
| Type of healthcare facility (n = 2,042) | Hospital | 1, 448 (70.9%) |
| | Health center | 505 (24.7%) |
| | Clinic | 39 (1.9%) |
| | Health post | 50 (2.5%) |
| Facility location (n = 2,042) | Urban | 1,396 (68.4%) |
| | Rural | 646 (31.6%) |
| Facility ownership (n = 2,042) | Public | 1,865 (91.3%) |
| | Other[d] | 177 (8.7%) |

a: *South Nations, Nationalities, and Peoples regions*

b: *Afar, Benshangul Gumuz and Harari, Dire Dawa, Gambella*

c: *Integrated emergency surgical officer, health extension worker level 3 or 4, or other clinical staff*

d: *Other governmental (military, prison, private for profit) and NGO (mission/faith-based, non-profit).*

**Table 2. Components of ANC received by participants by percentage, ESPA 2021–2022.**

| ANC component | Number completed | Percentage |
|---|---|---|
| Asked about client's age (n = 2041) | 630 | 30.9 |
| Asked about last menstrual period (n = 2,041) | 1,578 | 77.3 |
| Asked about number of prior pregnancies (n = 2,041) | 1,320 | 64.7 |
| Asked about current medications (2,041) | 314 | 15.4 |
| Asked vaginal bleeding (n = 2,011) | 965 | 48.0 |
| Asked Fever (n = 2,011) | 276 | 13.7 |
| Asked Headache/ blurred vision (n = 2,011) | 845 | 42.0 |
| Asked or examined legs/feet/hands for edema (n = 2,042) | 167 | 8.2 |
| Measured blood pressure (n = 2,042) | 1,623 | 79.5 |
| Measured weight (n = 2,042) | 1,546 | 75.7 |
| Examined conjunctiva/palms (n = 2,042) | 566 | 27.7 |
| Examined for swollen glands (n = 2,042) | 101 | 5.0 |
| Conducted breast examination (n = 2,042) | 67 | 3.3 |
| Palpated fundal height (n = 2,042) | 1,109 | 54.3 |
| Ordered/ tested haemoglobin (n = 2,039) | 1,377 | 67.5 |
| Ordered/ tested blood group (n = 2,041) | 1,463 | 71.7 |
| Ordered/performed urinalysis (n = 2,034) | 1,364 | 67.1 |
| Ordered/performed syphilis test (n = 2,038) | 1,391 | 68.3 |
| Ordered/ performed HIV test (n = 2,042) | 1,073 | 52.6 |
| Discussed healthy pregnancy (n = 2,039) | 796 | 39.0 |
| Discussed healthy diet (n = 2,039) | 967 | 47.4 |
| Discussed importance of at least 4 ANC (n = 2,039) | 387 | 19.0 |
| Discussed or provided iron, folic acid or both (n = 2,040) | 1,414 | 69.3 |
| Discussed or provided tetanus vaccine (n = 2,039) | 1,096 | 53.8 |

## Determinants of ANC quality

We conducted a bivariate GPR analysis to explore the relationship between the dependent variable and each of the chosen predictors. In the bivaraite GPR model, gestational age, educational status, area of residence, region, distance from healthcare facility, insurance coverage, sex of provider, facility type, and facility location were found to be significantly associated with the quality of ANC visit. However, after we adjusted for possible confounding factors in the multivariate GPR, gestational age, region, distance from healthcare facility, gender of the care provider, facility type, and facility location were found to be significantly associated with quality of ANC.

The quality of ANC was 1.088 times higher (IRR: 1.088; 95% CI: 1.011–1.171) among women from the Amhara region than among women from Addis Ababa. The quality of ANC was 1.081 times higher (IRR: 1.081; 95% CI: 1.002–1.166) among women from the SNNP regions than among women from Addis Ababa. Mothers who received the service from rural healthcare facilities were 3.6% times less likely (IRR: 0.964; 95% CI: 0.932–998) to receive high-quality ANC compared to those who received it from urban healthcare facilities.

Pregnant women receiving care from female providers were 1.054 times more likely to receive high-quality ANC than women receiving care from male providers (IRR: 1.054; 95% CI: 1.021–1.088). Participants receiving ANC from a nearby healthcare facility were 4.4% times less likely to receive quality ANC than thier counterparts (IRR: 0.956; 95% CI: 0.923–0.990).

When compared to participants who received care from hospitals, women who received care from clinics were 33.4% less likely (IRR: 0.666; 95% CI: 0.581–0.764) to receive quality

ANC. Women who received care from health posts were 18.3% less likely (IRR: 0.817; 95% CI: 0.732–0.911) to receive quality ANC. The number of weeks of pregnancy at which the pregnant women recieved their first ANC was negatively associated with quality of ANC, meaning that increased fetal gestational age at the first antenatal visit was associated with a decline in the quality of ANC (IRR: 0.994; 95% CI: 0.992–0.996) (Table 3).

## Discussion

We conducted this nationwide facility-based study to assess the quality of ANC and its determinants in healthcare facilities across Ethiopia. Investigating the existing healthcare service quality is crucial as it allows the implementation of strategies aimed at reducing maternal mortality in low- and middle-income countries such as Ethiopia [32]. Our findings show that pregnant women in Ethiopia received an average of 11 out of 24 ANC components recommended by the WHO during their first visit. Fetal gestational age at first antenatal visit, healthcare provider's gender, distance of a healthcare facility from a woman's home, and the healthcare facility location, region, and type were identified as determinants of quality ANC.

The WHO recommends that every pregnant woman receive comprehensive care during her first ANC visit [50]. However, this study found that, on average, pregnant women in Ethiopia received less than half of the ANC components recommended for their first visit. This finding is slightly lower than the report of an earlier study in Kenya, which showed that pregnant women received an average of 10.9 out of 16 ANC service components [51]. This discrepancy

**Table 3. Multiple generalized Poisson regression analysis of determinants of quality ANC.**

| Variable | Category | Unadjusted IRR (95% CI) | P | Adjusted IRR (95% CI) | P |
|---|---|---|---|---|---|
| Maternal educational status | No education | 0.979 (0.938–1.023) | 0.346 | 1.014 (0.961–1.070) | 0.601 |
| | Primary school | 0.939 (0.902–0.978) | 0.003 | 0.956 (0.911–1.003) | 0.069 |
| | Secondary school | 1.001 (0.960–1.043) | 0.972 | 1.010 (0.963–1.060) | 0.682 |
| | College & above | 1 | | 1 | |
| Facility location | Urban | 1 | | 1 | |
| | Rural | 0.953 (0.926–0.981) | 0.001 | 0.964 (0.932–0.998) | 0.015 |
| Region | Addis Ababa | 1 | | 1 | |
| | Amhara | 1.078 (1.014–1.145) | 0.016 | 1.088 (1.011–1.171) | 0.024 |
| | Oromia | 0.977 (0.922–1.036) | 0.428 | 1.017 (0.947–1.092) | 0.641 |
| | Somali | 0.964 (0.895–1.037) | 0.327 | 0.990 (0.903–1.086) | 0.835 |
| | SNNP[a] | 1.058 (0.995–1.126) | 0.072 | 1.081 (1.002–1.166) | 0.044 |
| | Sidama | 0.994 (0.923–1.071) | 0.877 | 1.030 (0.984–1.159) | 0.511 |
| | Other[b] | 1.032 (0.964–1.104) | 0.366 | 1.068 (0.984–1.159) | 0.113 |
| Healthcare facility nearest to home | Yes | 1 | | 1 | |
| | No | 0.950 (0.923–0.978) | 0.001 | 0.956 (0.923–0.990) | 0.011 |
| Covered by healthcare insurance | Yes | 1 | | 1 | |
| | No | 1.033 (1.001–1.066) | 0.041 | 1.022 (0.985–1.061) | 0.241 |
| Care provider's gender | Male | 1 | | 1 | |
| | Female | 1.044 (1.017–1.073) | 0.001 | 1.054 (1.021–1.088) | 0.001 |
| Healthcare facility type | Hospital | 1 | | 1 | |
| | Health center | 1.003 (0.973–1.034) | 0.840 | 0.977 (0.940–1.016) | 0.244 |
| | Clinic | 0.677 (0.603–0.760) | <0.001 | 0.666(0.581–0.764) | <0.001 |
| | Health post | 0.818 (0.745–0.898) | <0.001 | 0.817 (0.732–0.911) | <0.001 |
| Gestational age | N/A | 0.994 (0.992–0.995) | <0.001 | 0.994 (0.992–0.996) | <0.001 |

could be due to differences in healthcare facilities' readiness to deliver the service, including the availability of trained personnel and essential equipment among the countries, as indicated by other studies [36,52].

Failing to provide certain preventive ANC components, such as health education and counseling on healthy pregnancy, proper nutrition, and the importance of attending all visits, could hinder the majority of women in this study from preventing malnutrition during pregnancy and experiencing poor birth outcomes [53,54]. Similarly, failure to collect essential information during the first ANC visit, such as maternal age, last menstrual period, and signs of pregnancy-related risks from the majority of pregnant women, is concerning, as this information is crucial for identifying those who need additional follow-up and interventions, as well as for providing appropriate care [41]. Preservice training, supervision, and ongoing education for ANC providers are essential for delivering the service in accordance with the standards [55].

The likelihood of pregnant women receiving quality ANC decreased as the fetal gestational age at which they initiated the service increased. This finding agrees with other research from Ghana [56]. This may be because, as the gestational age at the first antenatal contact increases, there might be reduced emphasis on certain components of ANC, assuming that the pregnancy is progressing well. It has been shown that pregnant women who receive their first ANC check-up after three months of conception are less likely to receive the full complement of WHO-recommended ANC [57]. However, this finding contradicts a previous study conducted in Ethiopia showing that women who initiated ANC after four months of pregnancy were more likely to receive quality ANC compared to those who started their care earlier [58]. The disparity may stem from differences in study participants and data collection methods. The previous study involved pregnant women with repeated visits and used interviews, while our study observed the actual service provided during the first ANC visit. Enhancing women education and encouraging partner involvement in pregnancy care could be effective strategies for promoting earlier ANC visits in Ethiopia [59].

We found that pregnant women are less likely to receive quality ANC from rural healthcare facilities compared to those who receive it from urban facilities. This is supported by evidence from a Demographic and Health Survey conducted in Nepal where pregnant women residing in urban areas had the highest chance of receiving quality ANC [60]. The possible reason for the lower quality of ANC among women who received the service from rural healthcare facilities might be shortages of essential resources for the service. Scarcity of essential ANC resources, such as adequate financial support, infrastructure, and skilled healthcare providers, is quite common in low- to middle-income countries, including Ethiopia, and most healthcare facilities do not meet the recommended standards [61–63]. However, disparity in their availability between rural and urban areas is another challenge. A study by Defar A et al. in Ethiopia revealed that rural healthcare facilities had fewer essential resources compared to urban ones [36]. Furthermore, the challenges of retaining healthcare professionals in rural and remote areas [64] could exacerbate the situation, further limiting access to quality ANC for women in rural areas.

We also found that pregnant women using healthcare facilities in the Amhara and SNNP regions experienced better quality ANC compared to those receiving it in Addis Ababa. While healthcare facilities in Addis Ababa are better equipped with resources for ANC than those in Amhara and SNNP regions [65], there is evidence that availability of resources does not guarantee provision of quality ANC [66]. A reason for this difference could be that there are lower levels of healthcare provider adherence to ANC guidelines in Addis Ababa than in Amhara [67], and higher caseloads in Addis Ababa compared to the two regions [68]. Higher case loads and failure of healthcare providers to adhere to national guidelines would compromise the quality of care provided [58,69]. This implies that complete adherence to the

recommended guidelines during the first ANC plays a significant role in reducing maternal and neonatal complications, including deaths [70,71].

In line with studies conducted in Kenya and Nepal [51], our study showed that participants who received care in clinics and health posts experienced lower quality ANC compared to those who received care in hospitals. The disparities in the availability of essential ANC resources across facilities, often referred to as tracer items, could be a contributing factor. Tracer items are the essential minimum required for a functional healthcare facility to deliver a service [60,72]. According to a prior study in Ethiopia by Defar et al., the mean availability of tracer items, such as trained personnel for ANC, and supplies and drugs necessary for quality ANC, scored less than six out of ten. This availability was notably lower in health posts and clinics than in hospitals [36].

The quality of ANC for pregnant women was also influenced by the gender of healthcare providers. Pregnant women received better quality ANC from female healthcare providers than from males. Studies indicate that the gender of healthcare providers may influence the quality of healthcare, including ANC [73–75]. One possible reason is that female healthcare providers may have a better understanding of the unique needs and challenges faced by pregnant women, as they may have firsthand experience with pregnancy themselves. Another possible explanation is that female healthcare providers may be more comfortable discussing sensitive issues and providing more comprehensive care than their male counterparts [76,77]. These care attributes could improve the provider–client relationship, which has the potential to predict the quality of healthcare [78]. Although we understand that further studies might be needed to gather more information on the effects of healthcare providers' gender on the quality of ANC, we suggest that institutions teaching healthcare providers incorporate strategies to improve provider–client communication [77].

The fact that clients often do not use their nearby healthcare facilities remains a major challenge in providing quality maternal healthcare in Ethiopia [79]. This is despite encouragement for pregnant women to seek antenatal care from nearby healthcare facilities [80]. There is evidence that many women in developing countries bypass the facilities closest to their home because of subjective perceptions of poor service quality [81]. However, our study showed that pregnant women who bypassed nearby healthcare facilities were less likely to receive quality ANC than those using the nearest one. This might be due to the fact that bypassing can lead to overcrowding at the facility where care is sought, while the bypassed facility is underutilized, resulting in differences in providing quality ANC [82].

## Strengths and limitations of the study

Our study has several strengths. We used the most recent nationally representative healthcare facility survey data. The survey was conducted rigorously, employing standardized tools and trained data collectors. It also employed a standardized checklist that is widely recognized as the gold standard for evaluating the actual service provided [31]. However, it is important to consider some limitations of this study. The possibility of the Hawthorne effect, in which healthcare providers change their behavior because they are aware of being observed [83], could have influenced the study's results, potentially leading to an overestimation of the quality of care. The exclusion of certain populations, such as those from the Tigray region and women not attending their first ANC visit may limit the generalizability of the findings across Ethiopia. Moreover, the use of secondary data inherently limits the study to the ANC components available in the dataset, and its observational design restricts the ability to establish causality. Future research could overcome these limitations by including a broader sample and utilizing methodologies that support causal inference.

## Conclusion and recommendation

Our study indicates that, on average, pregnant women in Ethiopia receive less than half of the ANC components recommended by the WHO for their first ANC visit. Given the existing higher maternal and child health morbidity and mortality in Ethiopia [23], future intervention programs aimed at improving ANC quality in Ethiopia should encourage women to initiate ANC early and utilize nearby facilities, support healthcare providers in delivering standardized services through preservice training, supervision, and ongoing education, and ensure the availability and proper utilization of necessary ANC resources at all facilities. Moreover, future qualitative research will be important to explore how the gender of healthcare providers, particularly how it influences communication and rapport with patients during ANC visits, affects ANC quality and to identify related barriers.

## Supporting information

**S1 Table. Descriptions of the independent variables, ESPA 2021–2022.**
(DOCX)

## Acknowledgments

We would like to acknowledge the University of New South Wales for supporting this study. We also extend our appreciation to the MEASURE DHS Program and ICF International for granting us permission to utilize the ESPA data. We would also like to express our gratitude for support provided by SuperScript Writing and Editing.

## Author Contributions

**Conceptualization:** Addisu Alehegn Alemu, Alec Welsh, Marjan Khajehei.

**Data curation:** Addisu Alehegn Alemu.

**Formal analysis:** Addisu Alehegn Alemu.

**Methodology:** Addisu Alehegn Alemu, Marjan Khajehei.

**Software:** Addisu Alehegn Alemu.

**Supervision:** Alec Welsh, Theodros Getachew, Marjan Khajehei.

**Visualization:** Addisu Alehegn Alemu.

**Writing – original draft:** Addisu Alehegn Alemu.

**Writing – review & editing:** Addisu Alehegn Alemu, Alec Welsh, Theodros Getachew, Marjan Khajehei.

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
