## [Decision Letter · Decision Letter 0]

27 Aug 2024

PONE-D-24-30395Assessment of antenatal care quality in Ethiopia: Facility based study using service provision assessment dataPLOS ONE

Dear Dr. Alemu,

Thank you for submitting your manuscript to PLOS ONE. After careful consideration, we feel that it has merit but does not fully meet PLOS ONE’s publication criteria as it currently stands. Therefore, we invite you to submit a revised version of the manuscript that addresses the points raised during the review process.

Overall, this manuscript presents an important topic and employs appropriate methods for its analysis. However, major revisions are required before being acceptable for publication. In your revision, please consider my suggestions as well as the suggestions provided by Reviewer s for further consideration. 

We look forward to receiving your revised manuscript.

Kind regards,

Muhammad Haroon Stanikzai

Academic Editor

PLOS ONE

Journal Requirements:

Additional Editor Comments:

Overall, this manuscript presents an important topic and employs appropriate methods for its analysis. However, major revisions are required before being acceptable for publication. In your revision, please consider my suggestions as well as the suggestions provided by Reviewer s for further consideration.

1- Please proofread the whole article for language correction (Last line in abstract does not have a punctuation).

2- It has been suggested that 99% of maternal deaths in LMICs, including Sub-Saharan Africa, can be prevented with quality ANC [10, 11]. Please check your citations. I could not find this claim it the mentioned citations. Please use appropriate citations or remove this sentence.

3- Discussion: Recommendations for each of the key observations they have made.

4- I propose the authors use, consult, and add the following reference in the manuscript that assess ANC quality in another LMIC.

Stanikzai, M.H., Tawfiq, E., Jafari, M. et al. Contents of antenatal care services in Afghanistan: findings from the national health survey 2018. BMC Public Health 23, 2469(2023). https://doi.org/10.1186/s12889-023-17411-y

5- Please revise your tables according tho the journal style.

6- Please add line numbers in the revised manuscript.

Reviewers' comments:

Reviewer's Responses to Questions

**Comments to the Author**

1. Is the manuscript technically sound, and do the data support the conclusions?

Reviewer #1: Yes

Reviewer #2: Partly

2. Has the statistical analysis been performed appropriately and rigorously? 

Reviewer #1: Yes

Reviewer #2: Yes

3. Have the authors made all data underlying the findings in their manuscript fully available?

Reviewer #1: Yes

Reviewer #2: Yes

4. Is the manuscript presented in an intelligible fashion and written in standard English?

Reviewer #1: Yes

Reviewer #2: Yes

5. Review Comments to the Author

Reviewer #1: Dear Editors and Authors,

Thank you for giving me the opportunity to review the manuscript, titled “Assessment of antenatal care quality in Ethiopia: Facility based study using service provision assessment data”

The authors of the study chose a very important topic related to maternal and child health.

The study aimed to assess the quality of ANC and its associated factors in Ethiopia. The authors used data the 2021/2022 Ethiopian Service Provision Assessment survey and used data from pregnant women during their first ANC consultation. The authors assessed and summed 24 components of ANC components (ranged 0-24), with higher score indicating a better ANC quality. They employed and fitted a poisson regression model to examine predictors of ANC quality.

The authors found that the mean ANC quality was 11 (SD 3.8), with blood pressure measurement the most commonly performed ANC service, and breast examination for cancer screening the least performed component of ANC. They found that when ANC was provided by female health workers the ANC quality was higher, and that the ANC quality was lower in rural health facilities, and for women with increased gestational age, and for women who received care from non-nearby health facility.

My assessment of the manuscript is that it is well written and is well organized. The findings from this study have the potential to impact health interventions and policy to improve neonatal, children, and women’s health. However, the authors need to address the following two issues before the paper can be considered for publication.

1. In table 3, the column showing regression coefficients (SEs) is not necessary. The authors may opt to revise table 3 which should present IRR (95%CI) and p-values for unadjusted IRRs, and adjusted IRRs (which are the current IRRs in table 3).

2. The flow chart shows that out of 4355 women, data from 2042 women were analysed and 2313 women were excluded. In the discussion section as part of the limitations of the study, the authors may opt to discuss the extent of the possible selection bias because of the exclusion of 2313 women. It would be useful to provide some descriptive statistics similar to the one in table 1, as supplementary material, on the 2313 women who were excluded from this study.

Reviewer #2: First and foremost, thank you for the opportunity to review your manuscript. This study has significant potential to impact antenatal care quality improvement on a large scale in Ethiopia. I trust that the review process will contribute to further enhancing the manuscript and increasing its likelihood of reader understanding in this context.

The paper is well-constructed and presents a comprehensive study with several notable strengths, including a well-crafted title and clear organization. However, it would benefit from the inclusion of a conceptual framework to more effectively link the variables. Additionally, the claims should be more clearly positioned within the context of existing literature, supported by relevant citations. The feedback provided aims to assist in refining your work and enhancing its impact within the field.

To follow the comments go to file attached.

6. PLOS authors have the option to publish the peer review history of their article (what does this mean?). If published, this will include your full peer review and any attached files.

Reviewer #1: **Yes: **Dr Essa Tawfiq

Reviewer #2: No

---

## [Author Response · Author response to Decision Letter 0]

23 Sep 2024

Dear Editor and Reviewers,

First, we would like to express our gratitude to the journal editor and reviewers for their constructive feedback. We have addressed all the concerns raised and have thoroughly revised the manuscript. We believe it is now in a much-improved state. Below, you will find our detailed, point-by-point response. If you have any further concerns, please feel free to contact the corresponding author. Our point-by-point response is attached as a separate file.

Kind regards,

---

## [Decision Letter · Decision Letter 1]

7 Oct 2024

PONE-D-24-30395R1Assessment of antenatal care quality in Ethiopia: Facility based study using service provision assessment dataPLOS ONE

Dear Dr. Alemu,

Thank you for submitting your manuscript to PLOS ONE. After careful consideration, we feel that it has merit but does not fully meet PLOS ONE’s publication criteria as it currently stands. Therefore, we invite you to submit a revised version of the manuscript that addresses the points raised during the review process.

Thank you for addressing the initial comments provided by the reviewers. Based on the revised manuscript, the reviewers have requested that some minor comments still need to be addressed.

We look forward to receiving your revised manuscript.

Kind regards,

Muhammad Haroon Stanikzai

Academic Editor

PLOS ONE

Journal Requirements:

Additional Editor Comments:

Please make sure your tables align with PLOS ONE requirements.

Reviewers' comments:

Reviewer's Responses to Questions

**Comments to the Author**

1. If the authors have adequately addressed your comments raised in a previous round of review and you feel that this manuscript is now acceptable for publication, you may indicate that here to bypass the “Comments to the Author” section, enter your conflict of interest statement in the “Confidential to Editor” section, and submit your "Accept" recommendation.

Reviewer #1: All comments have been addressed

Reviewer #2: All comments have been addressed

2. Is the manuscript technically sound, and do the data support the conclusions?

Reviewer #1: Yes

Reviewer #2: Yes

3. Has the statistical analysis been performed appropriately and rigorously? 

Reviewer #1: Yes

Reviewer #2: Yes

4. Have the authors made all data underlying the findings in their manuscript fully available?

Reviewer #1: Yes

Reviewer #2: Yes

5. Is the manuscript presented in an intelligible fashion and written in standard English?

Reviewer #1: Yes

Reviewer #2: Yes

6. Review Comments to the Author

Reviewer #1: Dear Authors,

I thank you for addressing the comments I made in my previous review of your paper. Please also adhere to all requirements of PLOS ONE.

Reviewer #2: Thank you to the authors and editors for the opportunity to review the manuscript titled, “Assessment of antenatal care quality in Ethiopia: Facility-based study using service provision assessment data.” This is an important study on a critical topic related to maternal and child health, especially considering its large-scale scope.

Most of the previous comments have been addressed, and the manuscript has significantly improved. However, there are a few areas that still require refinement:

Abstract:

In the conclusion, the phrase “Thus, efforts are needed to ensure that quality ANC is provided for every pregnant woman in Ethiopia” could be further improved by providing more specific guidance for policymakers and authorities. It would be helpful to include concise recommendations based on the study's findings, focusing on how to improve ANC quality in the Ethiopian context in line with the study’s aim of assessing healthcare facility capacity.

Introduction:

This section has improved compared to the previous version. However, the authors should refine the conceptual framework of the ESPA to better align with ANC quality assessment as informed by DHS survey data. This would help readers better understand the scope of the study.

Methods:

The constructs in this section have been improved significantly.

Results:

No additional comments.

Discussion:

The discussion needs to better connect and justify the study’s findings. For instance, in line 272, the statement “suboptimal facility readiness, primarily caused by shortages of trained staff and equipment” should be revisited to ensure clarity and better integration with the overall discussion. Additionally, the subsequent sentences regarding WHO recommendations (lines 273-277) do not clearly link to the prior argument. These points should be reworded or modified to create a smoother flow and ensure coherence in presenting the study's implications.

In lines 278-287, the discussion on the frequency and quality of the first ANC visit needs further refinement. The authors should better articulate why and how to increase the number of pregnant women receiving their first ANC check-up before three months of gestation, using insights from the observed results in Table 1 (Characteristics of the study participants).

For example, factors such as "Partner’s involvement" could be highlighted as a potential strategy to encourage earlier ANC visits. Additionally, Table 2 presents critical data, such as:

Discussed healthy pregnancy: 39.0%

Discussed healthy diet: 47.4%

Discussed importance of at least 4 ANC visits: 19.0%

The relatively low percentages across these indicators underscore the need for improved ANC quality. The discussion should address these gaps and suggest specific interventions, such as better communication during consultations or enhanced health education, to promote early and frequent ANC attendance.

Strengthening this section will improve the manuscript’s clarity on how to address the barriers to early ANC and ensure alignment with the study's findings.

In line 307, the authors mention that "there is evidence that availability of resources does not guarantee provision of quality ANC [59]." However, this statement seems disconnected from the previous assertion in line 272, where it is suggested that "The low ANC content observed in our study could be due to suboptimal facility readiness, primarily caused by shortages of trained staff and equipment [45]."

To improve coherence, the authors should clarify the relationship between resource availability and ANC quality. If resource availability alone is not sufficient for ensuring quality care, the conclusion should focus more on other factors—such as the provider's gender or the proper use of resources—that contribute to the low quality of ANC. The authors should also provide clearer evidence or examples from their findings to support this argument, ensuring alignment between both statements. This refinement will strengthen the discussion and offer a more nuanced understanding of the factors influencing ANC quality.

Additionally, the conclusion seem so long , it should be more concise and clear.

7. PLOS authors have the option to publish the peer review history of their article (what does this mean?). If published, this will include your full peer review and any attached files.

Reviewer #1: No

Reviewer #2: No

---

## [Author Response · Author response to Decision Letter 1]

14 Oct 2024

Dear Editor and Reviewers,

First, we would like to express our gratitude to the journal editor and the reviewer for their feedback in improving our paper. We have addressed all the concerns, and we believe it is now in a much-improved state. Below, you will find our detailed point-by-point response. If you have any further concerns, please feel free to contact the corresponding author

---

## [Decision Letter · Decision Letter 2]

28 Oct 2024

Assessment of antenatal care quality in Ethiopia: Facility based study using service provision assessment data

PONE-D-24-30395R2

Dear Dr. Alemu,

We’re pleased to inform you that your manuscript has been judged scientifically suitable for publication and will be formally accepted for publication once it meets all outstanding technical requirements.

Kind regards,

Muhammad Haroon Stanikzai

Academic Editor

PLOS ONE

Additional Editor Comments (optional):

Reviewers' comments:

Reviewer's Responses to Questions

**Comments to the Author**

1. If the authors have adequately addressed your comments raised in a previous round of review and you feel that this manuscript is now acceptable for publication, you may indicate that here to bypass the “Comments to the Author” section, enter your conflict of interest statement in the “Confidential to Editor” section, and submit your "Accept" recommendation.

Reviewer #2: All comments have been addressed

2. Is the manuscript technically sound, and do the data support the conclusions?

Reviewer #2: Yes

3. Has the statistical analysis been performed appropriately and rigorously? 

Reviewer #2: Yes

4. Have the authors made all data underlying the findings in their manuscript fully available?

Reviewer #2: Yes

5. Is the manuscript presented in an intelligible fashion and written in standard English?

Reviewer #2: Yes

6. Review Comments to the Author

Reviewer #2: Thank you to the authors and editors for the opportunity to review the manuscript titled, “Assessment of antenatal care quality in Ethiopia: Facility-based study using service provision assessment data.” This is a valuable and timely study on a critical topic within maternal and child health, particularly given the large-scale scope of the data analyzed. The authors have comprehensively addressed all previously suggested revisions, resulting in an improved and well-structured manuscript to meet The journal criteria that

The study provides original research findings.

The results are novel and have not been published elsewhere.

The methodologies, including experiments and statistical analyses, are technically sound and thoroughly explained.

The conclusions are clearly stated and well-supported by the data.

The manuscript is well-written and easy to understand, meeting standard English language requirements.

The research maintains ethical standards and integrity throughout.

The study follows appropriate reporting guidelines, and data availability meets community standards.

So, I have no further comment.

7. PLOS authors have the option to publish the peer review history of their article (what does this mean?). If published, this will include your full peer review and any attached files.

Reviewer #2: No

---

## [Editor Report · Acceptance letter]

4 Nov 2024

PONE-D-24-30395R2 

PLOS ONE

Dear Dr. Alemu, 

I'm pleased to inform you that your manuscript has been deemed suitable for publication in PLOS ONE. Congratulations! Your manuscript is now being handed over to our production team.

Kind regards, 

on behalf of

Dr. Muhammad Haroon Stanikzai 

Academic Editor

PLOS ONE